# Optimizing Thermoelectric Performance of Tellurium via Doping with Antimony and Selenium

**DOI:** 10.3390/molecules28217287

**Published:** 2023-10-26

**Authors:** Manman Yang, Mengxiang Yang, Yimin Li, Yuqi Chen, Yuling Song, Jin Jia, Taichao Su

**Affiliations:** 1School of Electronic Engineering, Huainan Normal University, Huainan 232038, China; 2College of Science, Guilin University of Technology, Guilin 541004, China; 3School of Mechanical and Electrical Engineering, Nanyang Normal University, Nanyang 473061, China; 4Key Laboratory of Spin Electron and Nanomaterials of Anhui Higher Education Institutes, Suzhou University, Suzhou 234000, China

**Keywords:** thermoelectrics, tellurium, phonon thermal conductivity, high pressure and high temperature

## Abstract

Forming solid solutions is one of the most effective strategies to suppress the thermal conductivity of thermoelectric materials. However, the accompanying increase in impurity ion scattering usually results in an undesirable loss in hall mobility, negatively impacting the electrical transport properties. In this work, a tellurium–selenium (Te-Se) solid solution with trace antimony (Sb) doping was synthesized via the high pressure and high temperature method. It was found that slight Se doping into the Te sites not only had no impact on the hall mobility and carrier concentration, but also enhanced the density-of-state effective mass of Sb_0.003_Te_0.997_, leading to an enhanced power factor near room temperature. Additionally, the presence of Se doping caused a significant reduction in the phonon thermal conductivity of Te due to fluctuations in the mass and strain field. The lowest phonon thermal conductivity was as low as ~0.42 Wm^−1^K^−1^ at 600 K for Sb_0.003_Se_0.025_Te_0.972_, which approached the theoretical minimum value of Te (~0.28 Wm^−1^K^−1^). The effects of Se doping suppressed thermal conductivity, while Sb doping enhanced the power factor, resulting in a larger *ZT* of ~0.94 at 600 K. Moreover, these findings demonstrate that Sb and Se doping can effectively modulate the electrical and thermal transport properties of Te in a synergistic manner, leading to a significant increase in the average *ZT* across a wide temperature range.

## 1. Introduction

The use of thermoelectric (TE) materials has been recognized as a viable and environmentally friendly solution to address the global energy crisis [1,2,3,4,5]. The efficacy of TE devices in converting heat into electricity is primarily influenced by the dimensionless figure of merit, denoted as *ZT* = *S*^2^*T*/*ρκ*, where *S*, *ρ*, and *T* represent the Seebeck coefficient, electrical resistivity, and absolute temperature, respectively. Additionally, *κ* is the total thermal conductivity, comprising both electronic (*κ*_e_) and phonon (*κ*_ph_) thermal conductivities [6,7,8,9]. The optimization of *ZT* is constrained by the significant trade-off correlation observed between *S*, *ρ*, and *κ* as predicted by the Boltzmann transport theories and the Wiedemann–Franz relationship [10]. Consequently, achieving performance optimization becomes challenging due to the interdependence of these parameters [11,12,13]. In addition to possessing high *ZT* values, TE materials should also exhibit environmentally friendly characteristics and consist of non-toxic elements [14,15,16,17,18,19,20].

For an extended period, advanced TE materials have primarily consisted of compounds such as Bi_2_Te_3_ [21,22], PbTe [23], Mg_2_Si [24,25], half-Heuslers [26,27], and oxides [28]. Nevertheless, a significant drawback of these materials is their susceptibility to precipitation, segregation, and volatilization during the application process, making it challenging to achieve uniformity in the preparation process. Consequently, there is a pressing need to identify TE materials with simpler compositions. In 2014, elemental tellurium (Te) was found to exhibit promising characteristics as a promising TE material due to its relatively high band degeneracy (*N*_v_ = 4) near the valence band maximum (VBM), resulting in a large density-of-state (DOS) effective mass *m**_DOS_ and reaching a peak Seebeck coefficient (*S*) value of 450 μVK^−1^ at a hole concentration around 10^17^ cm^−3^, with an average value ranging from 200 to 250 μVK^−1^ at room temperature [29]. Recent studies have shown that the thermoelectric performance of the *p*-type Te-based system can be improved through the introduction of dopants such as arsenic (As) [30,31], antimony (Sb) [32], and bismuth (Bi) [33], which serve to modulate the carrier concentration in the Te substrate. The highest *ZT* value up to 1.0 at 650 K was successfully achieved in As-doped Te with a carrier density of 2.65 × 10^19^ cm^−3^ [30]. Previous studies have indicated that doping with non-equivalent elements is a fundamental and direct approach to enhance the thermoelectric performance of Te. However, the improvement in the electrical transport properties inevitably leads to a negative impact on the thermal performance. If the thermal conductivity can be mitigated while maintaining excellent electrical performance, the thermoelectric efficiency of Te can be significantly enhanced. According to our previous work, an extremal low thermal conductivity can be obtained in a Te-Se solid solution [34]. However, a large amount of Se alloying induces pronounced carrier scattering, thereby diminishing the hall mobility and adversely affecting the power factor. According to prior research, the application of high pressure has been found to induce abundant defects in the crystal structure, thereby facilitating the reduction in *κ*_ph_. Furthermore, the utilization of high-pressure technology has the potential to significantly decrease the preparation time from several days to a mere 30 min [35].

In this work, we investigated the effect of Sb doping and slight Se alloying on the thermoelectric properties of Te (Sb_0.003_Se*_x_*Te_0.997−*x*_, *x* = 0–0.05). Due to the strong phonon scattering from the mass and strain fluctuations, a remarkably low phonon thermal conductivity of ~0.42 Wm^−1^K^−1^ was achieved at 600 K, which was in close proximity to the theoretical minimum. Furthermore, we found that the formation of Te-Se solid solutions did not significantly affect the electrical properties of Sb_0.003_Te_0.997_ when the Se doping content was below 3.75 at.%. The low thermal conductivity and optimized power factor led to a high *ZT* of ~0.94 at 600 K for Sb_0.003_Se_0.025_Te_0.972_. This work provides significant insights into the optimization of thermoelectric properties through strategic doping and alloying. The findings suggest that the careful modulation of the material composition can lead to a balance between the thermal and electrical properties, ultimately enhancing the thermoelectric performance. The results establish a promising foundation for the development of highly efficient thermoelectric materials.

## 2. Results

Figure 1a provides the X-ray Diffraction (XRD) patterns of the Sb_0.003_Se*_x_*Te_0.997−*x*_ (*x* = 0–0.05) samples. In these patterns, all peaks align perfectly with the trigonal structure of Te, conforming to the P3121 space group. Importantly, no discernible impurity phase is detected within the sensitivity limits of the measurements for all of the samples under consideration. This observation attests to the purity of the synthesized compounds, indicating that the synthesis process is well controlled and free of extraneous elements. The inset in Figure 1a illustrates a magnified view of the main peaks. Here, a gradual shift towards higher angles is seen with the increasing Se concentrations. This trend signifies the successful integration of Se into the Te matrix. It also points towards a contraction of the lattice, a consequence of the smaller atomic radius of Se (1.16 Å) compared to that of Te (1.36 Å). Figure 1b presents the lattice parameter (*a*) of the Sb_0.003_Se*_x_*Te_0.997−*x*_ samples, as determined via the Rietveld refinement method using the General Structure Analysis System (GSAS) program. As the Se content increases, a corresponding decrease in the lattice parameters is observed. This change is aligned with the shift of the peaks towards higher angles that were previously noted in the XRD patterns, and it is consistent with Vegard’s law, which states that the lattice parameter of an alloy will linearly change with the concentration in a solid solution. These results strongly suggest that Te is being substituted by Se, leading to the formation of a solid solution. This conclusion corroborates the existing literature [36] and underscores the potential of Se alloying for tailoring Te’s structural properties. The findings provide critical insights into the crystallographic changes induced by Se alloying and its role in tuning the material’s thermoelectric performance.

Figure 2 presents a comprehensive visual depiction of the sectional morphology of the Sb_0.003_Se*_x_*Te_0.997−*x*_ compounds, where *x* varies as 0, 0.0125, 0.025, and 0.05. These compounds are synthesized by employing the high pressure and high temperature (HPHT) methodology, a technique renowned for its capability to generate materials with unique properties. In the figure, we can observe a clear pattern in the grain size of the synthesized samples. The Se-free sample, that is, when *x* = 0, exhibits a sizable grain structure with a measurement exceeding 20 µm, as shown in Figure 2a. This large grain size is indicative of the material’s characteristics in the absence of the Se alloy. Contrastingly, as we introduce the Se into the SbTe compound (increasing *x* value), a discernible shift in the grain structure is noted. The average grain size begins to decrease progressively, as visualized in Figure 2b–d. This trend is particularly interesting as it suggests a correlation between the Se concentration and the morphological characteristics of the synthesized compound. In a striking comparison, the Se-alloyed samples (where *x* > 0) present significantly smaller grain sizes, often less than 5 µm. This is a stark contrast to the Se-free samples and introduces a denser configuration of grains, leading to a considerable increase in the number of grain boundaries. This morphological modification, ushered by the introduction of the Se alloy, is expected to offer consequential impacts on the thermal properties of the resultant compound. The abundance of grain boundaries is theorized to lower the thermal conductivity (*κ*). This is because the increased grain boundaries serve as scattering centers for heat-carrying phonons, resulting in a reduction in the thermal conductivity. Hence, this research provides valuable insights into the structural and thermal properties of Sb_0.003_Se*_x_*Te_0.997−*x*_ compounds, highlighting the important role of the Se alloy in controlling these properties.

Figure 3 plots the electrical transport properties of pristine Te and Sb_0.003_Se*_x_*Te_0.997−*x*_ (*x* = 0–0.05) samples. As depicted in Figure 3a, the *ρ* of all samples increases with the increasing temperature (except for pristine Te), indicating degenerate semiconducting behavior. Furthermore, the *ρ* of Sb_0.003_Se*_x_*Te_0.997−*x*_ remains relatively constant for *x* ≤ 0.025, but experiences a sharp increase with the increasing Se content. Specifically, the *ρ* rises from ∼32 μΩm for Sb_0.003_Te_0.997_ to ∼89 μΩm for Sb_0.003_Se_0.05_Te_0.947_ at 300 K. Based on the Hall measurement results (Figure 3b), the observed increase in *ρ* at *x* > 0.025 can be primarily attributed to the decreased carrier concentration (*n*_H_) and mobility (*μ*_H_). This decrease is caused by the introduction of Se alloying, which induces variations in the band structure and the carrier scattering [37]. Notably, the returned *ρ* of the Se-alloyed sample remains significantly lower than that of the pristine Te, as Sb is utilized to adjust the *n*_H_. Figure 3c displays the temperature dependence of the Seebeck coefficient (*S*) for the Sb_0.003_Se*_x_*Te_0.997−*x*_ samples. The results indicate that all samples synthesized by HTHP exhibit a positive Seebeck coefficient (*S*), suggesting a predominance of hole (*p*-type) conduction. Moreover, the addition of Se to the matrix significantly enhances the Seebeck coefficient (*S*) value, with the sample alloyed with 5 at.% Se displaying a Seebeck coefficient (*S*) value of 213.8 μVK^−1^ at 300 K compared to 116.9 μVK^−1^ for the Sb_0.003_Te_0.997_ sample. However, for samples with *x* > 0.025, the Seebeck coefficient (*S*) value decreases with the increasing temperature beyond 500 K, indicating the occurrence of bipolar diffusion. The temperature dependence of the power factor (*PF*) is depicted in Figure 3d. It can be observed that the *PF* does not experience significant deterioration when the Se content remains below 2.5 at.%. Although the *PF* of samples with *x* > 0.025 deteriorate compared to the pristine Te_0.997_Sb_0.003_, it is still noticeably higher than that of pristine Te.

To further shed light on the electrical transport mechanism in the Sb_0.003_Se*_x_*Te_0.997−*x*_ (*x* = 0–0.05) samples, we employed the single parabolic band (SPB) model [38] to estimate the *m**_DOS_, as depicted in Figure 4a. The *m**_DOS_ of the Se-doped samples exhibits a slight increase, leading to an effective enhancement in the Seebeck coefficient. This also means that the band structure was altered after Se doping. However, it is worth noting that the substantially increased *m**_DOS_ in the highly doped sample (*x*~0.05) also hinders the carrier transport, as shown in Figure 4b. To elucidate the underlying mechanisms, we performed comprehensive electronic band structure calculations for pure Te and Se, as depicted in Figure 4c,d. While both materials exhibit generally similar band structures due to their comparable valence electron configurations, notable distinctions arise in the vicinity of the Valence Band Maximum (VBM), which plays a crucial role in determining the thermoelectric performance. Specifically, the VBM of Te is located near the H point and is characterized by a smaller effective mass. Conversely, the VBM of Se is situated at the L point and exhibits a larger effective mass. This higher effective mass contributes to a steeper density of states, thereby positively impacting the Seebeck coefficient. These computational results indicate that Se doping is advantageous for enhancing the effective mass, aligning well with our initial experimental observations.

As we know, achieving an extremely lower *κ* value and understanding the underlying phonon transport mechanism are extremely significant to the improvement of *ZT* for thermoelectric materials. Figure 5a plots the composition-dependent *κ* as a function of the temperature (*T*), revealing a nearly identical temperature dependence. The *κ* values exhibit a pronounced decrease with the increasing temperature, following the *T*^−1^ trend, suggesting the prevalence of phonon scattering through the Umklapp process. The *κ* decreases from 1.33 Wm^−1^K^−1^ for the pristine Te to 0.83 Wm^−1^K^−1^ for the sample alloyed with 5 at.% Se and doped with 0.3 at.% Sb. This represents reductions of approximately 22% and 38% compared to the Se-free sample (Sb_0.003_Te_0.997_) and pristine Te at 300 K, respectively. It is well known that *κ* is composed of two components, *κ*_e_ and *κ*_ph_. The value of *κ*_e_ can be determined using the Wiedemann–Franz law, *κ*_e_ = *LT*/*ρ*, where the Lorenz number (*L*) is estimated using the equation, *L* = 1.5 + exp[−|*S*|/116], in this study [39]. The *κ*_ph_ is obtained by subtracting the electronic component from *κ* as shown in Figure 5b. Obviously, the *κ* and *κ*_ph_ show a similar trend with Se doping and the increasing temperature. A significantly suppressed *κ*_ph_, reaching ~0.42 Wm^−1^K^−1^ at 600 K, is observed in the Sb_0.003_Se_0.025_Te_0.9753_ sample, approaching the amorphous limit of ~0.28 Wm^−1^K^−1^ calculated via Cahill’s model [40]. This reduction in *κ*_ph_ is nearly 50% lower than that of pristine Te at 600 K. Therefore, the decrease in *κ* is strongly correlated with the decrease in *κ*_ph_ for Sb_0.003_Se*_x_*Te_0.997−*x*_. Importantly, the *κ*_ph_ value of this work is considerably lower compared to that of Te(As) [30] and Te(Sb) [41] prepared using the conventional melt quench technique. It is worth noting that all the TE properties were assessed at an ambient pressure, indicating that the characteristics of the samples under a high pressure can be retained at an ambient pressure. This demonstrates that the HTHP-prepared samples possess residual stress, which aids in capturing the character, including the low conductance activation energy of the TE materials. The effect of a high pressure on the TE performance in this work was proven.

Figure 6 delves into the thermoelectric performance of pristine Te and the Sb_0.003_Se*_x_*Te_0.997−*x*_ (*x* = 0–0.05) samples, providing a detailed analysis of the key metrics related to thermoelectric efficiency. Figure 6a specifically exhibits the temperature dependence of the dimensionless thermoelectric figure of merit (*ZT*) for the aforementioned samples. In the field of thermoelectrics, *ZT* is a crucial metric that combines both the electrical and thermal properties of a material. It is noteworthy that the peak *ZT* value achieved is approximately 0.94 at a temperature of 600 K, as shown by the Sb_0.003_Se_0.025_Te_0.9753_ sample. This impressive performance can be attributed to the significant reduction in the thermal conductivity (*κ*), coupled with a relatively stationary power factor (*PF*), which suggests a commendable balance between electrical and thermal properties. Moving on to Figure 6b, the average *ZT* values (*ZT*_ave_) of the Sb_0.003_Se*_x_*Te_0.997−*x*_ (*x* = 0–0.05) samples, in the temperature range of 300 to 600 K, are summarized. The optimization of the density of states effective mass (*m**_DOS_) and a decrease in the phonon thermal conductivity (*κ*_ph_) contribute to the peak average *ZT* (*ZT*_ave_) of approximately 0.53 for Sb_0.003_Se_0.0125_Te_0.9845_. It is paramount to achieve a high *ZT*_ave_ across an extensive temperature range to enhance thermoelectric conversion efficiency, a crucial factor in thermoelectric applications. In pursuit of this goal, calculations were carried out to ascertain the thermoelectric power generation efficiency of *p*-type legs composed of Sb_0.003_Se*_x_*Te_0.997−*x*_ (*x* = 0–0.05) samples [42]. As per the conditions set for these calculations, the temperatures of the hot and cold sides of the legs were maintained at a steady 273/600 K, and neither thermal nor electrical losses were factored in. The outcomes of these calculations indicated a significant enhancement in efficiency, with an approximate 26% increase for the *x* = 0.0125 sample in comparison to the Se-free sample. These findings underscore the potential of defect engineering in achieving an optimal thermoelectric performance. The results highlight the delicate balance between carrier transport and phonon scattering, emphasizing the importance of the careful modulation of material defects to optimize these interactions and enhance the overall thermoelectric performance.

## 3. Experimental Procedure

### 3.1. Sample Preparation

The preparation of the samples involved a precise and controlled process to ensure the synthesis of the target compounds, namely pristine Te and Sb_0.003_Se*_x_*Te_0.997−*x*_, where *x* = 0, 0.0125, 0.025, 0.0375, and 0.05. High-purity materials were used as the starting ingredients: Te (99.99%), Sb (99.999%), and Se (99.95%). The proportions of these elements were carefully measured to align with the stoichiometric requirements of the targeted compounds. The first step in the preparation process was to thoroughly mix these elemental powders. This task was accomplished using a planetary ball mill, a device renowned for its ability to achieve highly homogeneous mixtures. To protect the materials from unwanted reactions with atmospheric gases, the milling process was conducted under an inert Ar gas atmosphere. Absolute alcohol was also employed to facilitate the milling process. Following the milling, the resultant mixture of Te, Sb, and Se elements was dried and then transformed into a cylindrical shape, preparing it for the subsequent high-pressure synthesis process. This shaping process helps to achieve uniform pressure and temperature conditions during the synthesis. The cylindrical samples were then subjected to a high-pressure synthesis process using a cubic anvil high-pressure apparatus (ZN-460, China). This process involved conditions of 4 GPa of pressure and a temperature of 1000 K, maintained for a duration of 30 min. After synthesis, the samples were rapidly quenched to room temperature within a span of 2 min to ensure a sharp transition and to preserve the high-temperature phase. The obtained ingots were then slowly heated to a temperature of 600 K over a span of 5 h, and then held at this temperature for an additional hour. This slow heating and soaking process helps to relieve any internal stresses in the material and to homogenize the sample. The ingots were then cut and polished, preparing them for subsequent structural and property characterizations. This meticulous preparation process ensures the synthesis of highly pure and structurally consistent samples, thereby providing a reliable basis for the subsequent analysis and evaluation of their thermoelectric properties.

### 3.2. Physical Measurements

To understand the characteristics of the synthesized materials, a series of comprehensive physical measurements were conducted. The phase structure of all samples was discerned through X-ray diffraction (XRD) analysis, employing Cu-Kα radiation (λ = 1.5406 Å) on a Rigaku SmartLabSE system, a high-performance XRD platform. This technique allows for the determination of crystallographic structures and the identification of any impurity phases. Microstructure morphology, providing insights into the grain size and distribution, was investigated using Scanning Electron Microscopy (SEM) on a Carl Zeiss Sigma 500 VP. SEM provides high-resolution images that reveal the surface features and structural details of the samples. The Hall effect, which provides valuable information on the type and concentration of charge carriers, was measured using the van der Pauw method facilitated by a Lake Shore 8400 Hall setup. The Seebeck coefficient (*S*), a measure of the thermoelectric power, and electrical resistivity(*ρ*), an indicator of charge transport efficiency, were simultaneously measured in the temperature range of 300 to 600 K. The measurements were carried out using the CTA-3S apparatus (Cryoall, China) with a temperature increment of 30 K and a heating rate of 5 K min^−1^. The thermal conductivity (*κ*), a critical parameter dictating the heat transport property of a material, was computed using the equation *κ* = *DC*_P_*ρ*. In this equation, *D* refers to the thermal diffusivity coefficient obtained through the laser flash method on a Netzsch LFA457 instrument. The parameter *ρ* denotes the pellet density measured via the Archimedes method. *C*_P_ signifies the specific heat capacity, which was estimated based on the Dulong–Petit law, under the assumption that it is not significantly influenced by temperature. The uncertainties associated with these measurements were within acceptable limits: ±5–7% for *κ* and *ρ*, and ±5% for *S*. These measurements provide a holistic understanding of the synthesized materials’ physical properties, contributing to our understanding of their thermoelectric performances.

### 3.3. Computational Method

The computational investigations into the properties of Te and Se were carried out using the projector-augmented wave (PAW) method, a highly revered technique for efficient and accurate quantum mechanical calculations. This method was implemented via the Vienna Ab initio Simulation Package (VASP), a widely recognized computational toolset for atomic-scale materials modeling [43]. In our computations, the exchange-correlation potential was treated using the Perdew–Burke–Ernzerhof (PBE) functional [44]. This functional, which is based on the generalized gradient approximation (GGA), is popular for its reliability and performance in predicting the properties of a wide array of materials. Further, we established an energy cutoff for the plane-wave basis set at 350 eV. This relatively high cutoff energy ensures the precision of the simulations by incorporating a sufficient number of plane waves in the calculations. To secure the convergence of our simulations, we required the total energy differences between consecutive steps to be less than 1 × 10^−6^ eV. This strict criterion ensures a high level of accuracy in the computation of the electronic structure. For the geometry optimizations, we employed a Monkhorst–Pack uniform *k*-point spacing of 0.15 Å^−1^. This parameter is crucial to adequately sample the Brillouin zone, and hence, to accurately determine the electronic properties of the material. Furthermore, the relaxation of the atomic positions was carried out until the Hellmann–Feynman force on each atom was less than 0.01 eV Å^−1^. This stringent force criterion ensures that the final atomic positions are at or very near their minimum-energy configurations. In conclusion, the computational methodology adopted in this study is designed to achieve a fine balance between the computational efficiency and the precise calculation of material properties. The results derived from these calculations provide valuable insights into the behaviors of Te and Se at the atomic level, helping to better understand the observed experimental results.

## 4. Conclusions

In this study, we successfully synthesized a series of samples comprising pristine Te and Sb_0.003_Se*_x_*Te_0.997−*x*_ (*x* = 0–0.05) using HPHT technology. Our aim was to enhance the TE properties of elemental Te through strategic Sb doping and Se alloying, and the results demonstrate the success of these techniques. The Sb doping proved instrumental in providing Te with superior electrical transport performance. The advantageous impact of Sb doping on the electrical properties lays the foundation for the high thermoelectric performance of these compounds. Meanwhile, the introduction of a modest amount of Se into the Te sites had a marked effect in suppressing the thermal conductivity of Sb_0.003_Te_0.997_. Given the disparities in mass and atomic radius between Se and Te, an extremely low phonon thermal conductivity of approximately 0.42 Wm^−1^K^−1^ was achieved at 600 K for the Sb_0.003_Se_0.025_Te_0.972_ composition. Interestingly, this significant reduction in thermal conductivity did not cause a detrimental effect on the carrier concentration and Hall mobility. This finding suggests that the slight Se alloying does not impinge on the power factor of the pristine Sb_0.003_Te_0.997_, preserving its exceptional electrical performance despite the addition of Se. Given the combination of reduced phonon thermal conductivity and a consistent power factor, the thermoelectric figure of merit (*ZT*) reaches a maximum value of approximately 0.94 at 600 K in the case of Sb_0.003_Se_0.025_Te_0.972_. This represents an appreciable increment of about 19% over the *ZT* value for Sb_0.003_Te_0.997_, highlighting the advantage of Se alloying in enhancing thermoelectric performance. In brief, our work demonstrates the viability of Sb doping and Se alloying as effective strategies for optimizing the thermoelectric properties of Te. The findings of this study pave the way for further investigations into the fine-tuning of material properties through doping and alloying, with potential implications for the design of high-performance thermoelectric materials.

## Figures and Tables

**Figure 1 molecules-28-07287-f001:**
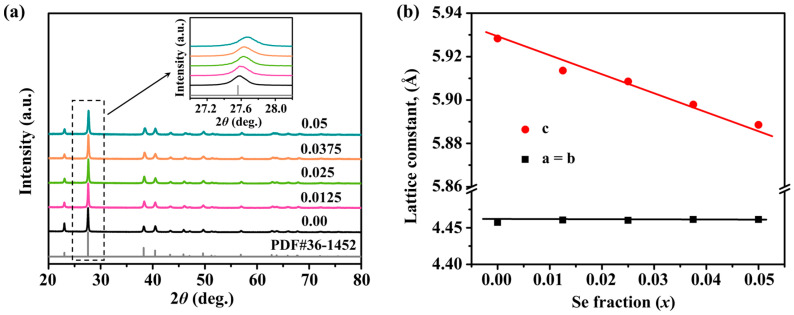
(**a**) Powder XRD patterns (inset is a close-up of the main peaks); (**b**) lattice parameters of Sb_0.003_Se*_x_*Te_0.997−*x*_ (*x* = 0–0.05) samples.

**Figure 2 molecules-28-07287-f002:**
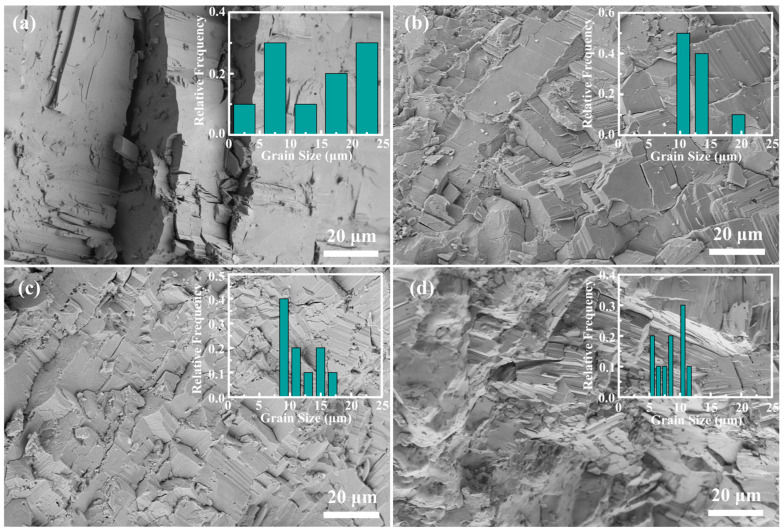
SEM images of the fresh fracture surface of Sb_0.003_Se*_x_*Te_0.997−*x*_ with (**a**) *x* = 0, (**b**) *x* = 0.0125, (**c**) *x* = 0.025, and (**d**) *x* = 0.05. Inset in (**a**–**d**): grain size distribution corresponding to (**a**–**d**).

**Figure 3 molecules-28-07287-f003:**
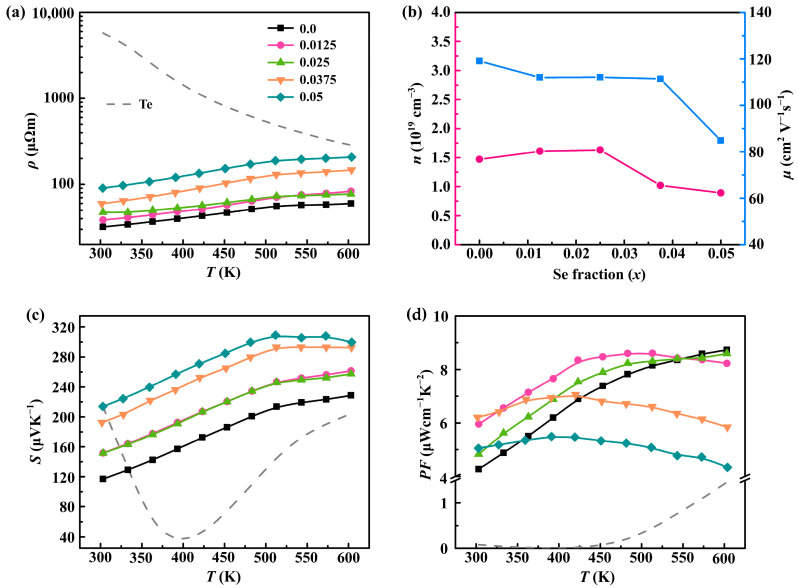
Temperature dependence of electrical transport properties: (**a**) electrical resistivity; (**b**) carrier concentration (pink line) and mobility (blue line); (**c**) Seebeck coefficient (*S*); and (**d**) power factor of both pristine Te and Sb_0.003_Se*_x_*Te_0.997−*x*_ (*x* = 0–0.05) samples.

**Figure 4 molecules-28-07287-f004:**
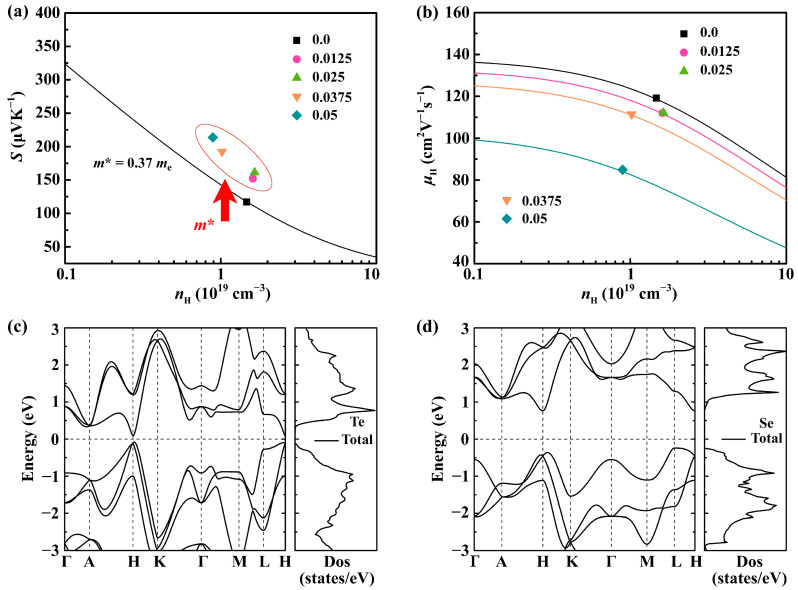
The relationship of (**a**) Seebeck coefficient (*S*); (**b**) Hall carrier mobility with carrier concentration of Sb_0.003_Se*_x_*Te_0.997−*x*_ (*x* = 0–0.05) samples; and the calculation results of electronic band structure for (**c**) Te and (**d**) Se, respectively.

**Figure 5 molecules-28-07287-f005:**
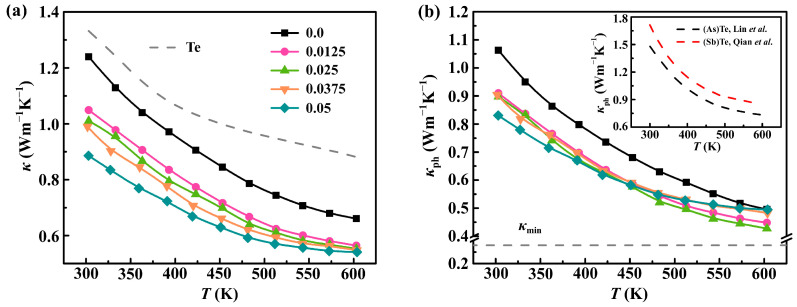
Temperature dependence of thermal transport properties: (**a**) total thermal conductivity; (**b**) phonon thermal conductivity for both pristine Te and Sb_0.003_Se*_x_*Te_0.997−*x*_ (*x* = 0–0.05) samples, inset in (**b**) reference data of Te-As [30] and Te-Sb [41] for comparison.

**Figure 6 molecules-28-07287-f006:**
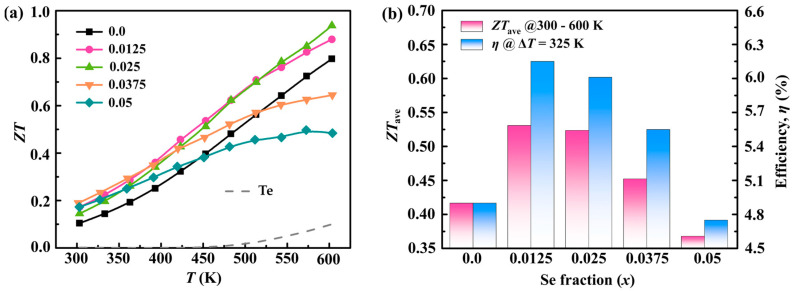
(**a**) Temperature-dependent *ZT* values; (**b**) *ZT*_ave_ values within the temperature range of 300–600 K and power generation efficiency for the pristine Te and Sb_0.003_Se*_x_*Te_0.997−*x*_ (*x* = 0–0.05) samples.

## Data Availability

Data are contained within the article.

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
