# Peer review of "Optimizing Thermoelectric Performance of Tellurium via Doping with Antimony and Selenium"

_molecules, 2023, doi:10.3390/molecules28217287_

Round 1

Reviewer 1 Report

Comments and Suggestions for Authors

Please see the enclosed file.

Author Response

Dear reviewer,

Thank you very much for your thoughtful consideration and for forwarding us your feedback on our manuscript. We have now carefully revised the manuscript in accordance with your suggestions. The key changes introduced in the revised version are highlighted in yellow, as follows:

Q1. You wrote: “…and a high Seebeck coefficient (S) [25]”. Please provide numerical value of S

Response: We have added the value of S for Ref [25] in the revised manuscript, as follows:

Lines 51-53 of pages 2: “In 2014, elemental tellurium (Te) was found to exhibit promising characteristics as a promising TE material due to its relatively high band degeneracy (Nv = 4) near the valence band maximum (VBM), resulting in a large density-of-state (DOS) effective mass m*DOS and reaching a peak Seebeck coefficient (S) value of 450 μVK-1 at a hole concentration around 1017 cm−3, with an average value ranging from 200 to 250 μVK-1 at room temperature [29].”

Q2. You wrote: “According to our previous work, an extrmal low thermal conductivity can be obtained in Te-Se solid solution [30].” There is nothing called “extrmal”.

Response: Thank you for the feedback. We have carefully re-examined the manuscript and made revisions to enhance the clarity and precision of the language throughout, with all changes highlighted in yellow.

Q3. You wrote: “However, a large amount of Se alloying induces pronounced carrier scattering, thereby 64 diminishing the hall mobility and adversely affecting the power factor.” This can be substantiated by providing a reference: https://doi.org/10.1016/j.jallcom.2019.01.105. This article discusses the importance of large ZTave in a wide temperature range

Response: Thanks for this suggestion and kind support. As recommended, we have calculated the TE conversion efficiency of the pristine Te and Sb0.003SexTe0.997-x (x = 0 - 0.05) samples according to the reference (J. Alloys Compd., 2019, 785, 862-870), with the results shown below:

Attached figure: Theoretical TE power generation efficiency for the pristine Te and Sb0.003SexTe0.997-x (x = 0 - 0.05) samples at different temperatures.

Furthermore, we have added additional data and accompanying description in the revised manuscript on pages 6-7, lines 189-199, as follows:

“Achieving a large ZTave in a wide temperature range is greatly desired for enhancing thermoelectric conversion efficiency. In light of this, we conducted calculations to determine the TE power generation efficiency of p-type legs composed of Sb0.003SexTe0.997-x (x = 0 - 0.05) samples [37]. In these calculations, the temperatures of the hot and cold sides of the legs were held constant at 273/600 K, with no thermal or electrical losses taken into account. The results revealed an approximate 26% increase in efficiency for x = 0.0125 when compared to the Se-free sample. The findings illustrate the viability of modulating defects to achieve optimal thermoelectric performance by balancing carrier transport and phonon scattering.

Figure 6. (a) Temperature-dependent ZT values; (b) ZTave values within the temperature range of 300 - 600 K and power generation efficiency for the pristine Te and Sb0.003SexTe0.997-x (x = 0 - 0.05) samples.

Q4. 2. Results appeared earlier than 3. Experimental Section. Usually Results and discussion section appear after Experimental Section.

Response: Thank you for the reminder. I understand your concern completely, but the journal has strict formatting requirements that manuscripts must adhere to.

Q5. The grain size of Se-free sample is over 20 μm (Figure 2a) and the average grain size gradually decreases with the increase of x value (Figure 2b-d). In comparison, the grain size of Se alloyed samples is significantly smaller (< 5 μm) than that of Se-free sample, resulting in an abundance of grain boundaries. What can be the plausible explanation for decreased grain size with an increase in x content?

Response: The smaller grain size of the Te-Se alloy was attributed to the lower diffusion rate and higher diffusion activation energy of the substitutional solid solution for grain growth, compared to pristine Te.

Q6. Grain size estimation using SEM images is not very accurate. Why X-ray peak size was not used to estimate the crystallite size (Sherrer formula)?

Response: When grain size is less than 100 nm, stress-induced peak broadening can be ignored compared to broadening from small grain size. In this case, the Scherrer formula is applicable. However, when grain size increases substantially (>100 nm), stress-induced broadening becomes more prominent. At this point, broadening from stress must be taken into account, and the Scherrer formula is no longer valid. Therefore, in this work, we did not employ the Scherrer formula to estimate crystallite size.

Q7. Specifically, why HPHT technology was used to fabricate the samples? Please explain in a few sentences

Response: HPHT technology is a novel materials preparation approach with several advantages for fabricating samples. Firstly, high pressure can lower the activation energy of chemical reactions, shortening the preparation cycle from several days to just 30 min—dramatically reducing the material’s synthesis time. Additionally, high pressure permits tuning of the material's bandgap, playing a unique role in optimizing electrical transport properties important for thermoelectric. Furthermore, high pressure introduces abundant defects in the crystal structure, aiding the reduction of thermal conductivity. These factors are critical to enhancing thermoelectric performance. In summary, HPHT is a beneficial method for property improvement through shorter timescales, structural modification, and defects engineering.

Furthermore, we have added some relevant details in the revised manuscript on page 2, lines 67-70, as follows:

“According to prior research, the application of high pressure has been found to induce abundant defects in the crystal structure, thereby facilitating the reduction of κph. Furthermore, the utilization of high-pressure technology has the potential to significantly decrease the preparation time from several days to a mere 30 min [35].”

Reviewer 2 Report

Comments and Suggestions for Authors

The paper is easy to read and already in good shape. 

There are only minor changes, which would improve the paper. 

- In fig 2: use the same x-axis for the histograms for the grain sizes in all 4 figures to make a comparison easier

-Put the electrical investigation before the thermal investigations, because for the phonon thermal conductivity one need to know the electrical thermal conductivity 

- Insert a legend as well for figure 5b

- insert a legend for figure 6b

- I would either put the experimental section as 2nd chapter or in the annex but not below the results

Comments on the Quality of English Language

no further comments

Author Response

Dear reviewer,

Thank you sincerely for your thoughtful consideration and feedback on our manuscript. We have now revised the manuscript carefully according to your suggestions. The key changes introduced in this revised version are highlighted in yellow, as follows:

Q1. In Fig. 2: use the same x-axis for the histograms for the grain sizes in all 4 figures to make a comparison easier

Response: According to your suggestion, we have now used a consistent x-axis scale across all inset histograms showing grain size distributions in Figure 2, as follows::

Figure 2. SEM images of the fresh fracture surface of Sb0.003SexTe0.997-x with (a) x = 0, (b) x = 0.0125, (c) x = 0.025, and (d) x = 0.05. Inset in (a-d): grain size distribution corresponding to (a-d).

Q2. Put the electrical investigation before the thermal investigations, because for the phonon thermal conductivity one need to know the electrical thermal conductivity

Response: In the revised manuscript, we have rearranged the order of the electrical and thermal property investigations, which are now presented between pages 3 - 6.

Q3. Insert a legend as well for Figure 5b

Response: We have added a legend and reordered Figure 5b in the revised manuscript on page 5, line 149 as follows:

Figure 4. The relationship of (a) Seebeck coefficient (S); (b) Hall carrier mobility with carrier concentration of Sb0.003SexTe0.997-x (x = 0 - 0.05) samples; the calculation results of electronic band structure for (c) Te and (d) Se, respectively.

Q4. insert a legend for Figure 6b

Response: In the revised manuscript on page 7, line 196, we have added a legend for Figure 6b as follows:

Figure 6. (a) Temperature-dependent ZT values; (b) ZTave values within the temperature range of 300 - 600 K and power generation efficiency for both pristine Te and Sb0.003SexTe0.997-x (x = 0 - 0.05) samples.

Q5. I would either put the experimental section as 2nd chapter or in the annex but not below the results

Response: Thank you for the feedback. However, as this journal has fixed guidelines for manuscript formatting and layout, we have structured the paper according to the publication's requirements.

Round 2

Reviewer 1 Report

Comments and Suggestions for Authors

Dear authors,

I have concluded reviewing the revised version of the manuscript. All the concerns/suggestions are addressed. I noticed a typo that can be easily fixed. This can be done while article is in the proof state. I therefore go ahead and recommend accepting the manuscript for publication.

Page 9, line 281, authors wrote:

" electrical conductivity (ρ),"

Should be electrical resistivity.

Best regards,

Nagaraj